# Transmission of Zearalenone, Deoxynivalenol, and Their Derivatives from Sows to Piglets during Lactation

**DOI:** 10.3390/toxins13010037

**Published:** 2021-01-06

**Authors:** Xandra Benthem de Grave, Janine Saltzmann, Julia Laurain, Maria A Rodriguez, Francesc Molist, Sven Dänicke, Regiane R Santos

**Affiliations:** 1Schothorst Feed Research, 8218 NA Lelystad, The Netherlands; XBenthemdeGrave@schothorst.nl (X.B.d.G.); FMolist@schothorst.nl (F.M.); 2Institute of Animal Nutrition, Friedrich-Loeffler-Institute (FLI), Federal Research Institute for Animal Health, D-38116 Brunswick, Germany; Janine.Saltzmann@fli.de (J.S.); Sven.Daenicke@fli.de (S.D.); 3Olmix, 56580 Brèhan, France; JLaurain@olmix.com (J.L.); mrodriguez@olmix.com (M.A.R.)

**Keywords:** zearalenone, metabolites, sows, piglets, milk, serum

## Abstract

Sows were fed naturally contaminated diets containing: (i) 100 ppb zearalenone (ZEN) one week before farrowing and during the lactation period (at 26 days), (ii) 100 ppb ZEN one week before farrowing and 300 ppb ZEN during the lactation period, or (iii) 300 ppb ZEN one week before farrowing and during the lactation period. All diets contained 250 ppb deoxynivalenol (DON). The highest levels of ZEN, α-ZEL, or β-ZEL were observed in the serum of sows fed 300 ppb ZEN before farrowing and during lactation. However, only α-ZEL was significantly increased in the colostrum and milk of these sows. Sows fed the 300 ppb ZEN during the complete trial presented a significant decrease in backfat thickness before farrowing. This effect was accompanied by a decrease in serum leptin levels. These sows also presented a decrease in estradiol levels and this effect was observed in their piglets exposed during lactation, which presented increased glucagon-like peptide 1, but no changes in serum levels of ZEN, α-ZEL, or β-ZEL. Although all sows were fed the same levels of DON, the serum levels of DON and de-epoxy-DON were increased only in the serum of piglets from the sows fed a diet with the highest ZEN levels during the whole experimental period. Moreover, these piglets presented gut inflammation, as indicated by significantly increased calprotectin levels in their serum.

## 1. Introduction

Zearalenone (ZEN) and deoxynivalenol (DON) are the most commonly found mycotoxins in food and feed, especially those based on corn and beet pulp [1,2]. It is well known that exposure to high concentrations of ZEN or DON results in their transmission into milk. In a previous study, we observed the negative impact of feeding sows diets contaminated with different ZEN levels (200 to 1000 ppb) during gestation and lactation, which resulted in impaired ovarian function of the piglets [3]. However, under field conditions, a contaminated corn-based diet usually contains a combination of ZEN and DON. Even following the EU guidance [4] value of these mycotoxins in the final diet, transmission into the colostrum and milk from sows and into their piglets may occur.

ZEN and its derivatives, e.g., α-ZEL and β-ZEL, have a conformation similar to that of estradiol, allowing them to bind estrogenic receptors, leading to an estrogenic action [5]. This shows the importance of measuring ZEN derivatives and estradiol levels in milk and serum. Besides this, some xenoestrogens affect the synthesis of other hormones and proteins. For example, genistein (a phytoestrogen present in soybeans) inhibits leptin synthesis [6]. Nevertheless, it is not clear if ZEN could cause the same effect. Leptin, mainly secreted by adipocytes, has several regulatory functions such as safeguarding normal mammary gland development, lactation, immune responses, appetite regulation, and energy balance [7]. Leptin also stimulates glucagon-like peptide 1 (GLP1) release, which is a gut-derived hormone playing a physiological role in satiety [8].

A short-term exposure study around parturition, i.e., the last week of gestation combined with a lactation period of 26 days, could be helpful for better understanding of the risks of piglets’ maternal exposure. To date, there is no information on the recovery of ZEN, DON, and their derivatives in the colostrum of sows. To evaluate the effects of ZEN, two different batches of beet pulp were used to prepare the diets, resulting in a diet containing 100 ppb ZEN, and another with 300 ppb ZEN. When preparing diets with different feedstuffs, a multi-mycotoxin contamination will rarely be avoided. Although the beet pulp used in the present study was not a source of DON, the combination of all the other feedstuffs to prepare the feed resulted in a final diet containing approximately 250 ppb DON. Although this level is far lower than the recommended one (900 ppb) [4] and can be regarded as background contamination of pig feed, some impact on the intestinal integrity was hypothesized, particularly in combination with the intended ZEN dosing regimen, since ZEN was also shown to affect intestinal integrity [9]. Therefore, serum markers of inflammation like calprotectin [10] and also the anti-inflammatory marker GLP1 [11] were selected to test this hypothesis. Furthermore, the serum levels of the tight junction protein zonula occludens-1 (ZO-1) could indicate the occurrence of gut leakage [12].

The aims of the present study were to determine (i) if ZEN, DON, and their derivatives can be recovered in the colostrum and the milk of sows fed naturally contaminated diets, (ii) if ZEN, DON, and their derivatives will reach blood circulation of suckling piglets, (iii) if ZEN at different levels (100 vs. 300 ppb) in the diet will affect performance parameters of lactating sows and suckling piglets, and (iv) if hormones and proteins are affected in sows and suckling piglets when ZEN and DON are present in sows’ diets.

## 2. Results

### 2.1. Sows and Litter Performance

In Table 1, the feed intake, body weight, and backfat development of the sows are shown. The experiment had two main ZEN levels in the diets (low: ~100 ppb also called LoZEN; 336 high: ~300 ppb also called HiZEN). Average feed intake during the whole experimental period (day 109 of gestation until weaning) was 5.4 kg/day. At start of the experiment sows had an average body weight of 269 kg.

No effect of the dietary treatments was observed regarding sows’ body weight gain or average daily feed intake. Although a significant decrease in the backfat thickness was observed at farrowing for the sows fed a HiZEN diet from day 109, this effect was not observed after farrowing. No effects of dietary treatment on litter size, litter weight, or individual piglet weight were observed (Table 2). On average, sows in the experiment had 15.7 total born (TB) piglets and 14.9 live born (LB) piglets with an average birth weight of 1395 grams (LB).

Litters were standardised between 24 and 48 h after birth at 12–14 piglets per litter resulting in an average of 13.3 piglets per sow. Dietary treatment did not affect litter/piglet pre-weaning growth nor did it affect pre-weaning mortality (Table 3). In this experiment, the piglets had an average weaning weight of 8.4 kg, which means an average growth of 258 g/d until weaning. No significant differences were observed.

### 2.2. Analysis of Biological Samples

#### 2.2.1. Mycotoxins in Colostrum and Milk

A significant increase of α-ZEL levels was observed in the colostrum of sows from T3, i.e., those fed a HiZEN diet from d109 until farrowing. When the milk was evaluated at weaning, the increase of α-ZEL levels was observed in sows from T2 and T3, i.e., those fed a HiZEN diet from farrowing to weaning or from d109 to weaning, respectively (Table 4). The transfers of ZEN from feed to colostrum in the ZEN form were of 0.004%, 0.003%, and 0.002% for T1, T2, and T3, respectively. The transfers of ZEN from feed to milk in the ZEN form were of 0.004%, 0.002%, and 0.002% for T1, T2, and T3, respectively. Likewise, the transfers of DON from feed to colostrum were low, being 0.03%, 0.01%, and 0.03% for T1, T2, and T3, respectively. The transfers of DON from feed to milk were of 0.04%, 0.05%, and 0.04% for T1, T2, and T3, respectively. These values did not differ significantly.

#### 2.2.2. Mycotoxins in Serum

At the start of the trial, no differences were observed in the serum levels of mycotoxins. However, at the end of the trial, i.e., day 26 of lactation, a significant increase was observed in the serum levels of ZEN, α-ZEL, and β-ZEL of sows fed diets T2 and T3 when compared with the control (T1). Although there were no significant differences in the levels of ZEN and its derivatives in the serum from piglets, the serum levels of DON and de-DON were significantly high in the piglets from T3 (Table 5).

#### 2.2.3. Serum Levels of Estradiol, Leptin, Glucagon-Like Peptide 1, Calprotectin, and Zona Occludens-1

No differences were observed when comparing the serum levels of sows at day 109 of gestation. At weaning, serum estradiol levels decreased significantly in sows that were fed a HiZEN diet from day 109 of gestation to weaning. The presence of HiZEN in the diet of the sows from day 109, or from farrowing up to weaning, resulted in a significant decrease in leptin serum levels. Regarding the piglets, serum levels of estradiol and GLP1 were significantly decreased in piglets from sows fed HiZEN diet from day 109 of gestation or from farrowing up to weaning. On the other hand, these piglets presented a significant increase in calprotectin serum levels (Table 6).

## 3. Discussion

To the best of our knowledge, this is the first study demonstrating the transmission of ZEN, DON, and their metabolites from sows to piglets during lactation, via colostrum and milk. Although there are some studies showing the detection of ZEN and its derivatives in milk from humans [13,14] and cattle [15], up to now, no information has been given for sows fed realistic doses of ZEN obtained from naturally contaminated diets. Regarding DON and de-DON, they have been previously identified in the colostrum and serum from sows [16]. In a recent study, Trevisi et al. [17] demonstrated the transfer of dietary DON to the colostrum of sows fed naturally contaminated diets containing this mycotoxin at levels ranging from 60 to 438 ppb.

The dietary exposure was sufficient to significantly increase the levels of α-ZEL in the colostrum from sows fed a diet containing 300 ppb ZEN when compared to a diet with 100 ppb ZEN. Zearalenone is rapidly metabolised by the gastrointestinal tract after ingestion, producing some metabolites that are even more toxic, such as α-ZEL [18]. When the milk was evaluated at weaning, a significant increase of α-ZEL levels was observed in sows fed a HiZEN diet from farrowing to weaning, regardless of the ZEN level before farrowing. DON levels in colostrum and milk were similar among the groups, since the dietary levels of DON were the same for all sows. Although dietary exposure did not affect sows’ performance, a significant decrease in backfat thickness was observed in the sows fed a HiZEN diet from the last week of gestation until farrowing. Changes in backfat thickness were corrected for the value at the start of the experiment, i.e., day 109 of gestation, to exclude the effect of parity or other individual characteristics of the sows. This effect was combined with a decrease in serum leptin levels at farrowing and after weaning. Backfat consists of water, collagen, and lipid, and is also a source of different hormones including leptin, which is positively correlated with backfat thickness. A decrease in backfat thickness represents backfat loss or mobilisation. Valros et al. [19] reported that the fat content in the milk of sows varies accordingly with backfat thickness. In the present study, no changes in milk nutrients (protein, fat, and lactose) were observed (data not shown). Decaluwé et al. [20], on the other hand, demonstrated that backfat thickness during late gestation predicts colostrum yield. Unfortunately, colostrum yield was not measured, and mycotoxin levels in the backfat were not assessed in the present study. Previously, Goyarts et al. [21] detected DON in the backfat of fattening pigs. Therefore, we cannot exclude the backfat as another source of mycotoxins to colostrum and milk.

At weaning, a significant decrease of estradiol levels was observed in the serum from sows fed the HiZEN diet during the whole experimental period. This observation differs from a previous study where ZEN did not affect plasma levels of estradiol in 25 kg gilts [22]. These authors found that estradiol levels in the samples were below the limits of sensitivity for all groups, including the control. Differently, Long and Diekman [23] also demonstrated that exposure of gilts to ZEN during gestation resulted in a decrease in the serum concentrations of estradiol. In the present study, such a decrease of estradiol was also observed in the suckling piglets. In the past, we have shown that ZEN exposure during lactation would result in a depletion of the pool of gametes present in the ovaries of female piglets due to the estradiol depletion caused by ZEN [3]. Therefore, although production performance was not impaired, the exposed female piglets should not enter reproduction programs. Serum levels of DON and de-DON in the sows were not affected by the diets.

The transmission of ZEN via the milk of sows was previously demonstrated in a study where two sows were fed a diet containing 40,000 ppb crystalline ZEN for a period of nine days starting from day 8 after farrowing [24]. The authors observed that suckling female piglets from these sows presented reddening and swelling of the vulva, and the most abundant ZEN derivative was β-ZEL. None of these morphological changes were observed in the present study and the most abundant compounds in milk were ZEN and α-ZEL. It is impossible to compare both datasets due to the huge differences in dietary ZEN levels, as well as the methods used for mycotoxins analysis. In a previous study, we showed the effects of the dietary exposure of sows to 200 ppb ZEN during gestation (115 days) and lactation (21 days) on their progeny [3]. In the present study, the sows were exposed on the short-term to ZEN and DON, i.e., during the last week of gestation and during the 26 days of lactation. Wipperman et al. [25] exposed sows to 4420 ppb DON during days 35 and 70 of gestation and performed histo-morphological and immunohistochemical analyses on 70-days-old porcine fetuses, but no pathologies were detected in the fetal organs. Likewise, Goyarts et al. [26] exposed sows to 4420 ppb DON and 48 ppb ZEN during days 35 and 70 of gestation and no negative impact was observed in the foetuses. Similarly to our study, no effects on performance were observed by the aforementioned authors.

In the present study, the exposure time to ZEN during gestation was very short (one week before farrowing). The transfer of DON and ZEN during the last third of gestation was previously demonstrated by the recovery of these mycotoxins and their derivatives in the serum, urine, and bile from neonate piglets [27]. Besides the fact that ZEN, α-ZEL, and β-ZEL were detected in the serum from the suckling piglets at weaning, the maternal exposure to ZEN and DON during the last week of gestation and lactation was sufficient to significantly increase the levels of DON and de-DON, together with a trend of increased α-ZEL level in the serum from piglets. The piglets that were exposed during the last week of gestation and during lactation also presented a significant increase in calprotectin serum levels, as well as a decrease of the serum GLP1 when compared to mycotoxin exposure during lactation only. The impact of the interaction of ZEN and DON on the reproductive performance of pigs has previously been described [28,29], whereas their impact on intestinal function and integrity still needs to be elucidated. It has been claimed that ZEN and its derivatives have an anti-inflammatory effect on enterocytes [9,30]. In rats, however, ZEN exposure has resulted in intestinal inflammation [31]. Furthermore, ZEN, β-ZEL, and especially α-ZEL can decrease the integrity of intestinal porcine epithelial cells (IPEC-1). Such anti-inflammatory effect is not present anymore, probably in the presence of DON, a pro-inflammatory mycotoxin. When tested in combination, DON and ZEN are usually offered to the target animals at high levels [32,33,34] and no information is given on the role of ZEN in increasing DON absorption. It is known that intestines also contain estrogenic receptors and that xenoestrogens may affect gut function and integrity. For instance, exposure to ZEN via oral gavage resulted in altered intestinal microflora and intestinal inflammation in mice [35]. Nevertheless, more studies are needed to explain the interaction between these two mycotoxins commonly present in the swine diet. Increased serum calprotectin levels indicate that inflammation is taking place, as observed in the present study. Usually, faecal calprotectin is used as a marker for inflammation in the gastrointestinal system, once its concentration in faeces is about six times that in serum/plasma [36]. Due to the availability of serum, and because it was not possible to collect faeces individually from the piglets, we performed the serum analysis only. All calprotectin measured levels were low (~13 to 20 ng/mL) compared to levels reached in infection studies performed on pigs, where the animals were exposed to *Escherichia coli*, and calprotectin reached levels higher than 100 ng/mL in plasma [37]. Therefore, the serum calprotectin levels measured in the present study are not a sign of clinical inflammation, but demonstrate that an inflammatory process was taking place. Concomitantly with the increase in calprotectin levels, the serum levels of GLP1 were decreased. This gut hormone exerts an anti-inflammatory effect in the intestine [11], supporting the hypothesis that piglets from the sows fed HiZEN and DON diets were experiencing inflammation. This gut hormone is also released in response to the presence of nutrients in the intestine and, when released, it will inhibit food intake in humans [38]. Therefore, an increase in feed intake would be expected when GLP1 levels decrease. In the present study, a numerical increase in the intake of creep feed was observed in the piglets presenting a significant decrease in serum GLP1 levels. However, no increase in growth was observed, and one must bear in mind that the intake of creep feed only is not a robust parameter because it should be necessary to determine if piglets are drinking more and, sometimes, creep feed is lost from the feeder and not really eaten. No changes on the serum levels of ZO-1 were observed. This effect does not discard the possibility of increased gut permeability but indicates that a possible leakage was not critical enough to reach a clinical finding. As observed in the sows, piglets suckling from sows fed HiZEN diets presented a significant decrease in serum oestradiol levels.

In conclusion, it is possible to determine ZEN, DON, and their derivatives in the colostrum and milk from sows fed contaminated diets with mycotoxins at practical EU levels. Exposure to HiZEN resulted in a decrease in backfat thickness at farrowing and such an effect was correlated with a decrease in leptin serum levels in sows. Such an exposure also resulted in a decrease in the serum estradiol levels of sows and piglets. Although the transmission of mycotoxins did not affect the performance parameters of suckling piglets, an inflammatory process was taking place, and it was probably not specifically related to the intestine based on the findings on calprotectin and ZO-1 serum levels. When exposed to HiZEN during the last week of gestation and during lactation, the piglets presented increased DON and de-DON levels in their serum. However, this information is not sufficient to state that ZEN and its derivatives increased the absorption of this other mycotoxin. Besides, dietary levels of 3+15 Ac-DON and DON-3-G were also below the LOQ. Nonetheless, these levels may have still been high enough to increase the DON level in the intestine of suckling piglets. This short-term exposure will not affect pig production. Due to ethical concerns, the weaning period in Europe is increasing to at least 28 days. In practice, this will result on an averaged weaning period of 30 days and, consequently, increase the period of lactational exposure. Furthermore, pigs are useful models to evaluate human exposure to toxicants. In this case, awareness should be raised about the risks of infant exposure during lactation.

## 4. Materials and Methods

### 4.1. Animal Ethics Statement

This study was conducted according to the guidelines of the Animal and Human Welfare Codes/Laboratory practice codes in the Netherlands. The protocol was approved by the Ethics Review Committee: Body of Animal Welfare at SFR (AVD246002015280), approval date: 8 January 2019. 

### 4.2. Animals and Housing

A total of 15 sows (parity 1 to 6) in good health and normal body condition were used in this experiment. Sows had an average parity of three and parity was balanced across treatments as much as possible. During gestation, sows were housed in groups of approximately 150 sows and four feeding stations were available per group. Sows were fed a commercial diet (Appendix A) during gestation with marginal levels of mycotoxins. Sows were transferred from the gestation unit to the farrowing rooms at day 109 of gestation. The farrowing pens (0.60 × 2.50 m for sow; 2.25 × 2.50 m total surface) were equipped with a feeding bin and sows were able to fill the feeder by pushing a metal bar in the feeder. Sows did not have access to straw or other bedding material. Pens had a plastic slatted floor, including a heated section for piglets programmed to reach 40 °C at farrowing to 30 °C at three weeks after birth. The room temperature schedule was decreasing from 24 °C at farrowing to 20 °C at five days after farrowing. Artificial light was provided from 6:00 till 22:00 h. Cross fostering was done within 48 h after birth within a treatment and parity group to balance the litter size to 12–14 piglets. From day 14 after birth and until weaning, all the piglets received creep feed (Appendix A), which was not contaminated with mycotoxins.

### 4.3. Diets and Experimental Design

The diets were prepared with naturally contaminated feedstuffs and, as a consequence, the *Fusarium* mycotoxin DON was present in all of them at the same level (~250 ppb). As a ZEN source, two batches of sugar beet pulp were used in the present study. One had negligible contamination, while the other was highly contaminated with ZEN. All other main feedstuffs (corn, soybean meal, wheat, sunflower seed meal, soybean hulls, and clean beet pulp batch) were present at the same inclusion levels in all diets. As a result, the experiment had two main ZEN levels in the diets (low: ~100 ppb or LoZEN; high: ~300 ppb or HiZEN). The recommended maximum level of ZEN in sows diet is 250 ppb [4].

The experiment comprised in total three dietary treatments and five replicates per treatment, where each sow was a replicate. Treatments were randomly allocated to the sows. One week before farrowing, sows were moved to the lactation unit and the gestation diet was replaced by a lactation diet with low or high levels of ZEN. Dietary treatments and diet codes are summarised in Table 7. From day 14 after birth until weaning (day 26) all piglets received an uncontaminated commercial creep feed. Sows and piglets were monitored daily for abnormalities, such as abnormal behaviour, clinical signs of illness, and mortality throughout the experiment.

All diets were analysed in an independent and accredited (BELAC 057-TEST/ISO17025) laboratory (Primoris Holding, Gent, Belgium) via liquid chromatography with tandem mass spectrometry (LC-MS/MS).

This multi-mycotoxin analysis was applied confirming that ZEN was the main contaminant, followed by DON, while other mycotoxins were found at low to negligible levels. Importantly, although all diets also contained low levels of DON, this mycotoxin was at a constant level. Values of all detected mycotoxins in the diets are presented in Table 8.

### 4.4. Measurements

#### 4.4.1. Performance

Sows were individually weighed at day 109 of gestation, on the day of farrowing, and at weaning. Backfat thickness was also measured at these points in time. The sows’ feed intake was calculated by the difference in feed allowance and feed refusals during the experimental period. After this, the average feed intake and body weight gain during the whole experimental period (day 109 of gestation until weaning) were calculated. The gestation length and number of born (alive and still) piglets were also recorded.

Piglets were weighed at birth and at weaning. The intake of creep feed was also recorded from day 14 up to day 26. Average feed intake and growth were calculated. Mortality rate was recorded in sows and piglets.

#### 4.4.2. Levels of ZEN, DON and Their Derivatives in Colostrum, Milk, and Serum Samples

Serum samples of the sows were collected before starting the feeding trial at day 109 of gestation. Colostrum samples of the sows were collected at farrowing for mycotoxins analysis. At weaning (day 26 after farrowing), milk and serum samples were collected from the sows and serum samples were collected from 10 piglets per sow. The method of analysis has previously been validated and described [15,39]. In brief, the analysis was performed on a 4000 QTrap mass spectrometer equipped with an ESI source (Applied Biosystems, Darmstadt, Germany) and a 1200 series LC system (Agilent Technologies, Böblingen, Germany). The analytical column was a Pursuit XRs Ultra C18 column (100 × 2 mm, 2.8 μm; Agilent Technologies). A binary gradient of LC-MS grade water as eluent A and MeOH–ACN (70:30) as eluent B was used to separate ZEN, DON, and their metabolites. The ESI-MS/MS was performed in negative mode using a multiple reaction monitoring (MRM) technique. The serum samples were prepared as described by Brezina et al. [39]. A slightly modified sample preparation method was used for milk and colostrum [15]. LOD and LOQ values were calculated from low spiked milk and serum samples based on signal-to-noise ratios of 3:1 and 10:1 using the Analyst Software tool and the quantifier transition (Table 9). Samples were not corrected for recovery (Table 9), which was calculated as the ratio of the concentration obtained from the calibration curve and the known spiking level.

Transfer of ZEN and DON to colostrum and milk was calculated as previously described with a slight modification [21]. In the present study, instead of calculating the carry-over factor, the percentage of transfer was calculated as mycotoxin concentration in the colostrum or milk divided by the mycotoxin exposure via feed. The toxin exposure was calculated by multiplying the toxin concentration of the diet with the feed intake of each sow and dividing it by body weight [40] between the start of the trial and farrowing for the colostrum, and between farrowing and weaning for the milk.

#### 4.4.3. Serum Levels of Estradiol, Leptin, Glucagon-Like Peptide 1, Calprotectin, and Zona Occludens-1 in Sows and Piglets

Levels of estradiol (pg/mL), leptin (ng/mL), GLP1 (pg/mL), calprotectin (ng/mL), and ZO-1 (ng/mL) were measured using assay kits from MyBiosource Inc. (San Diego, CA, USA), coded as MBS282789, MBS703419, MBS943508, MBS033848, and MBS2608203, respectively. Absorbance was measured at a wavelength of 450 nm (plate reader Infinite^®^ 200 Pro, Tecan, Männedorf, Switzerland).

#### 4.4.4. Statistical Analysis

The experimental data were analysed with ANOVA (GenStat Version 20.0, 2020). Each sow was an experimental unit. Data collected from piglets (10 per sow) were used as a mean value per sow. Treatment means were compared by least significant difference (LSD). Values with *p* ≤ 0.05 were considered statistically significant. The following statistical model was used:*Yij* = µ + Blocki + Treatmentj + eij,(1)
in which:*Yij* = dependent variable,(2)
*µ* = overall mean,(3)
*Blocki* = block (i= replicate 1–5),(4)
*Treatmenti* = effect of treatment (j=1, 2, 3),(5)
*eij* = residual error,(6)

For all parameters parity was used as a covariate to minimise the effect of age. Furthermore, the body weight and backfat thickness of the sow at the start of the experiment was used as a covariate for body weight and backfat development. The percentage of stillborn piglets, individual piglet weight at birth, and coefficient of variance at birth were corrected for the number of total born piglets. Pre-weaning litter growth and pre-weaning mortality were corrected for the number of live born piglets and weaning age.

## Figures and Tables

**Table 1 toxins-13-00037-t001:** Effect of dietary treatments on average daily feed intake (ADFI), body weight (BW), BW gain (BWG), backfat (BF), and BF gain (BFG) ^1^.

	T1 (n = 5)	T2 (n = 5)	T3 (n = 5)	LSD	*p*-Value
ADFI (kg/d)					
d109-d14 of lactation	4.4	3.8	3.9	1.05	0.53
Farrowing-d14 of lactation	4.8	4.6	4.6	1.21	0.93
d109-weaning	5.1	4.8	4.8	1.25	0.84
Farrowing-weaning	5.5	5.5	5.4	1.37	0.98
BW development (kg)					
BW d109	267	271	269	48.3	0.99
BW farrowing	254	252	248	9.11	0.38
BW weaning	220	220	219	26.6	>0.99
BWG d109-farrowing	−15.4	−16.6	−21.2	9.11	0.38
BWG farrowing-weaning	−44.2	−32.5	−29.4	15.7	0.14
BWG d109-weaning	−48.7	−48.9	−50.2	26.6	>0.99
BF development (mm)					
BF farrowing	15.5 ^b^	15.7 ^b^	14.2 ^a^	1.24	0.05
BF weaning	13.2	12.5	12.5	1.26	0.46
BFG d109-farrowing	0.21 ^b^	0.48 ^b^	−1.08 ^a^	1.24	0.05
BFG farrowing-weaning	−2.31	−3.25	−1.64	1.65	0.12
BFG d109-weaning	−2.10	−2.77	−2.72	1.26	0.43

^a,b^ different superscripts indicate significant differences among treatments (*p* ≤ 0.05). ^1^ All values of body weight and backfat thickness were corrected for body weight and backfat thickness of the sow at the start of the experiment. T1: LoZEN from d109 gestation until d26 of lactation. T2: LoZEN from d109 gestation until farrowing and HiZEN from farrowing until d26 of lactation. T3: HiZEN from d109 gestation until d26 of lactation.

**Table 2 toxins-13-00037-t002:** Effect of dietary treatment on performance of the sow at farrowing.

	T1	T2	T3	LSD	*p*-Value
Gestation length (d)	114.8	116.0	114.2	2.08	0.20
Number of piglets (n)					
Total born	14.8	15.2	17.0	5.30	0.64
Live born (LB)	14.0	14.8	16.0	4.34	0.62
Still born	0.8	0.4	1.0	1.49	0.68
Birth weight (LB)					
Litter (kg)	19.7	21.8	18.6	4.96	0.38
Individual (kg) ^1^	1.4	1.5	1.4	0.41	0.83

^1^ Values are corrected for the number of total born piglets. T1: LoZEN from d109 gestation until d26 of lactation. T2: LoZEN from d109 gestation until farrowing and HiZEN from farrowing until d26 of lactation. T3: HiZEN from d109 gestation until d26 of lactation.

**Table 3 toxins-13-00037-t003:** Effect of dietary treatment on piglet performance.

	T1	T2	T3	LSD	*p*-Value
Weaning age (d)	27.2	26.0	27.0	2.21	0.47
Number of piglets					
Start ^1^	12.8	13.4	13.6	1.21	0.36
Weaning	11.6	13.0	12.8	1.87	0.25
Weaning weight					
Litter (kg) ^2^	107.0	107.8	99.8	17.0	0.51
Piglet (kg) ^2^	8.7	8.7	8.0	1.22	0.35
CV (%) ^2^	12.5	13.0	18.0	7.19	0.19
Growth					
Piglet (g/d) ^2^	265	265	244	38.7	0.38
Litter (kg/d) ^2^	3.3	3.3	3.0	0.54	0.45
Mortality					
Weaning ^1^ (%)	5.3	5.6	7.2	9.95	0.90
Creep feed intake					
Piglet (g/d)	16.1	21.6	26.8	18.11	0.44

^1^ After standardisation. ^2^ Values are corrected for the number of piglets at standardisation and weaning age. T1: LoZEN from d109 gestation until d26 of lactation. T2: LoZEN from d109 gestation until farrowing and HiZEN from farrowing until d26 of lactation. T3: HiZEN from d109 gestation until d26 of lactation.

**Table 4 toxins-13-00037-t004:** Effect of dietary treatment on mycotoxins levels (ng/mL) in colostrum and milk.

	T1	T2	T3	LSD	*p*-Value
*Colostrum*					
ZEN	0.053	0.048	0.075	0.0624	0.62
α-ZEL	0.094 ^a^	0.066 ^a^	0.218 ^b^	0.0793	<0.01
Deoxynivalenol DON	0.75	0.36	0.76	1.013	0.63
*Milk*					
ZEN	0.120	0.130	0.115	0.0823	0.93
α-ZEL	0.062 ^a^	0.176 ^b^	0.156 ^b^	0.0920	0.043
DON	2.98	3.19	2.37	1.932	0.64

^a,b^ Different superscripts indicate significant differences among treatments (*p* ≤ 0.05). β-ZEL, ZAN, α-ZAL, β-ZAL, and de-DON were not detected in the samples. T1: LoZEN from d109 gestation until d26 of lactation. T2: LoZEN from d109 gestation until farrowing and HiZEN from farrowing until d26 of lactation. T3: HiZEN from d109 gestation until d26 of lactation.

**Table 5 toxins-13-00037-t005:** Effect of treatments on the mycotoxin levels (ng/mL) in the serum from sows and piglets.

	T1	T2	T3	LSD	*p*-Value
Sows					
Day 109 of gestation					
ZEN	0.066	0.076	0.066	0.0443	0.84
α-ZEL	0.148	0.266	0.258	0.2960	0.64
β-ZEL	0.057	0.053	0.081	0.0378	0.27
DON	0.445	0.477	0.929	0.8210	0.11
de-DON	-	-	-	-	-
Day 26 weaning					
ZEN	0.43 ^a^	1.08 ^b^	1.11 ^b^	0.281	<0.001
α-ZEL	0.94 ^a^	3.09 ^b^	3.42 ^b^	0.984	<0.001
β-ZEL	0.23 ^a^	0.37 ^b^	0.53 ^c^	0.131	<0.01
DON	3.07	3.21	2.10	1.919	0.42
de-DON	0.11	0.53	0.15	0.543	0.22
Piglets at weaning					
ZEN	0.024	0.020	0.024	0.0232	0.91
α-ZEL	0.011	0.012	0.040	0.0279	0.08
β-ZEL	0.042	0.040	0.041	0.0035	0.59
DON	0.045 ^a^	0.045 ^a^	0.099 ^b^	0.0469	0.05
de-DON	0.040 ^a^	0.040 ^a^	0.058 ^b^	0.0160	0.05

^a–c^ Different superscripts indicate significant differences among treatments (*p* ≤ 0.05). ZAN, α-ZAL, β-ZAL were not detected in the samples. T1: LoZEN from d109 gestation until d26 of lactation. T2: LoZEN from d109 gestation until farrowing and HiZEN from farrowing until d26 of lactation. T3: HiZEN from d109 gestation until d26 of lactation.

**Table 6 toxins-13-00037-t006:** Effect of dietary treatment on the levels of estradiol, leptin, GLP1, calprotectin, and zona occludens-1 in the serum from sows and piglets.

	T1	T2	T3	LSD	*p*-Value
Sows					
Day 109 of gestation					
Estradiol (pg/mL) ^1^	>1000	>1000	>1000	*	*
Leptin (ng/mL)	2.0	1.7	1.5	0.43	0.11
GLP1 (pg/mL)	25.1	18.2	18.7	20.49	0.73
Calprotectin (ng/mL)	3.1	3.2	2.6	1.16	0.48
ZO-1 (ng/mL)	0.6	0.4	0.6	1.18	0.89
Day 26 weaning					
Estradiol (pg/mL)	31.2 ^b^	27.3 ^ab^	20.9 ^a^	7.79	0.04
Leptin (ng/mL)	1.6 ^b^	1.3 ^a^	1.0 ^a^	0.31	<0.01
GLP1 (pg/mL)	24.5	10.1	14.6	18.54	0.26
Calprotectin (ng/mL)	11.2	18.2	19.6	17.71	0.56
ZO-1 (ng/mL)	0.6	0.3	0.1	0.72	0.31
Piglets					
Day 26 weaning					
Estradiol (pg/mL)	77.1 ^b^	40.0 ^a^	45.2 ^a^	29.72	0.04
Leptin (ng/mL)	4.1	4.2	5.4	2.58	0.49
GLP1 (pg/mL)	17.5 ^c^	12.0 ^b^	5.1 ^a^	5.24	<0.001
Calprotectin (ng/mL)	13.4 ^a^	19.0 ^ab^	20.5 ^b^	5.77	<0.01
ZO-1 (ng/mL)	0.2	0.2	0.4	0.44	0.60

^a–c^ Different superscripts indicate significant differences among treatments (*p* ≤ 0.05). T1: LoZEN from d109 gestation until d26 of lactation. T2: LoZEN from d109 gestation until farrowing and HiZEN from farrowing until d26 of lactation. T3: HiZEN from d109 gestation until d26 of lactation. ^1^ Measured levels were above the maximum of quantification; * Not possible to perform statistical analysis.

**Table 7 toxins-13-00037-t007:** Experimental treatments.

Treatments	Day 109 of Gestation Till Farrowing	Farrowing to 26 Days of Lactation
1	LoZEN	LoZEN
2	LoZEN	HiZEN
3	HiZEN	HiZEN

All diets also contained ~250 ppb DON, regardless of the treatment. LoZEN: ~100 ppb; HiZEN: ~300 ppb.

**Table 8 toxins-13-00037-t008:** Multi-mycotoxins analyses of the diets (levels in ppb).

	Diets	
Mycotoxins (ppb)	LoZEN	HiZEN
Zearalenone	118	318
Deoxynivalenol	259	255
Fumonisin B1+B2	83.1	84.0
Alternariol	26.9	29.5
Alternariol ME	68.5	68.6
Beauvericin	19.8	27.7
Enniatin A1	3.5	-
Enniatin B	32.6	28.9
Enniatin B1	9.2	8.6

Below detection level in all diets: Aflatoxin B1, B2, G1, and G2, 3+15 Ac-DON, DON-3-G, Nivalenol, Ochratoxin A, T2 & HT2 Toxin, Diacetoxyscirpenol, Cytochalasine E, Sterigmatocystin, Alternariol ME, Citrinin, Roquefortine C, Enniatin A, A1, B and B1, Moniliformin.

**Table 9 toxins-13-00037-t009:** Limit of detection (LOD) and quantification (LOQ) from serum and milk samples and their corresponding recovery of ZEN, DON, and their metabolites.

	Serum			Colostrum/Milk		
	Recovery ± SD	LOD	LOQ	Recovery ± SD	LOD	LOQ
ZEN	105 ± 4	0.01	0.03	112 ± 2	0.04	0.12
α-ZEL	100 ± 4	0.02	0.08	94 ± 2	0.04	0.14
β-ZEL	92 ± 4	0.08	0.26	90 ± 4	0.10	0.34
ZAN	85 ± 4	0.07	0.24	102 ± 9	0.22	0.73
α-ZAL	97 ± 3	0.31	1.03	98 ± 3	0.26	0.88
β-ZAL	102 ± 4	0.69	2.31	103 ± 2	0.81	2.71
DON	108 ± 4	0.09	0.28	111 ± 3	0.11	0.36
de-DON	102 ± 7	0.08	0.26	108 ± 8	0.20	0.13
ZEN	105 ± 4	0.01	0.03	112 ± 2	0.04	0.12

## Data Availability

Data sharing not applicable.

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
