# Peer review of "Transmission of Zearalenone, Deoxynivalenol, and Their Derivatives from Sows to Piglets during Lactation"

_toxins, 2021, doi:10.3390/toxins13010037_

Round 1

Reviewer 1 Report

This paper is a significant contribution to the literature in the subject area of mycotoxicosis. The authors did an excellent job of presenting the information in a clear and concise manner. I would only suggest some minor edits (below):

line 34 please cite references for this statement. I would also clarify that although the transfer is well documented it is a small percentage. Therefore the importance of actually studying these low levels as opposed to just calling them unimportant as it is typically done in the field (present company included). In my humble opinion, this is a very important contribution of research in this paper. 

Line 66 at this point ZO-1 has not been previously defined. I only saw it defined in the list of definitions at the end. The authors define GLP-1 in the paragraph before I think it would be beneficial to define Z0-1 here. 

229-230 the reference should be included. I assuming it is the reference (1) in the references but it needs to be included here for clarity.

Line 410 P is missing from the word percentage

336-337 LowZen and HiZen are first defined in line 336-337 although they appear early in the publication line 88-89. I understand they are defined in the Abbreviations section but it is not clear when that section will be in the final version. In the sequence the paper is presented (Discussion before M&M), it would help clarify if it was defined early in the discussion. 

Reviewer 2 Report

The aims of the present study were to determine if ZEN, DON, and their deriva- 68 tives can be recovered in the colostrum and the milk of sows fed naturally contaminated 69 diets, ZEN, DON, and their derivatives will reach blood circulation of suckling piglets, 70 ZEN at different levels in the diet will affect performance parameters 71 of lactating sows and suckling piglets, and if hormones and proteins are affected in 72 sows and suckling piglets when ZEN and DON are present in sows’ diets.
The purpose of the work is clearly stated. The conclusions of the conducted research are clear and result from the obtained research results. The material used for the research is sufficient, the research methods have been selected appropriately. The arrangement of the figures is clear and presents the obtained results very well.
Discussing the results against the background of other authors is very detailed. The publications cited by the authors of the article are well selected. For the most part, the authors refer to the latest knowledge published in renowned scientific journals.

Author Response

We are very glad to read that the present paper brings clear information to the present reviewer, and we would like to acknowledge the compliments.

Reviewer 3 Report

Line 42: “ZEN and its derivatives, e.g., α-ZEL and -ZEL, have a conformation” → “ZEN and its derivatives, e.g., α-ZEL and β-ZEL, have a conformation”

Lines 114-119: Please explain how the transfers of ZEN and DON were calculated.

Table 4: The unit of mycotoxins’ level should be provided.

Lines 126-129 & Figure 1: If we do not know for sure the half-life of ZEN in sows’ body, we should not expect that ZEN and its derivative levels in sows’ milk were only affect by the dietary ZEN content during lactation period. Therefore, it should not be plotted in two groups. Three groups should be more appropriate.

Figure 1: There were 15 sows in the experiments. Please explain why there were only 14 dots in A and D.

Lines 150-152 & Figure 2: If we do not know for sure the half-life of ZEN in sows’ body, we should not expect that ZEN and its derivative levels in sows’ milk were only affect by the dietary ZEN content during lactation period. Therefore, it should not be plotted in two groups. Three groups should be more appropriate.

Figure 2: Please explain why there were not 15 dots in each sub-figure.

Lines 152-153: Please explain why no correlation between ZEN or α-ZEL levels in serum and milk was observed.

Table 6: Please explain what is day 0.

Table 6: Footnote for 1 and * should be added.

Lines 196-197: According to Materials and Methods, serum samples were taken only at weaning from sows. How to know there was a decrease of leptin at farrowing?

Lines 228-230: Please list the reference for the previous study.

Lines 295-297: Sows fed high level of ZEN did not showed increased absorption of DON. Only piglets of group 3 showed increased DON and its derivative. Group 2 did have a high level of ZEN intake in sows. Therefore, it is not a strong solid proof that ZEN increased DON absorption.

Round 2

Reviewer 3 Report

Manusrcipt has been carefully modified.